# Bootstrapped Physically-Primed Neural Networks for Robust T2 Distribution Estimation in Low-SNR Pancreatic MRI

**Hadas Ben Atya**[1] (ID)                                          HDS@CAMPUS.TECHNION.AC.IL
**Nicole Abramenkov**[1]                                  NICOLEA@CAMPUS.TECHNION.AC.IL
**Noa Mashiah**[1]                                NOA.MASHIAH@CAMPUS.TECHNION.AC.IL
**Luise Brock**[1,2]                                                 LUISE.BROCK@FAU.DE
**Daphna Link Sourani**[1,3]                               LDAPHNA@BM.TECHNION.AC.IL
**Ram Weiss**[4]                                      RAMW@RAMBAM.HEALTH.GOV.IL
**Moti Freiman**[1,3] (ID)                              MOTI.FREIMAN@TECHNION.AC.IL

[1] *Faculty of Biomedical Engineering, Technion – Israel Institute of Technology, Haifa, Israel*

[2] *Pattern Recognition Lab, Friedrich-Alexander-Universität, Erlangen, Germany.*

[3] *The May-Blum-Dahl MRI Research Center, Faculty of Biomedical Engineering, Technion- IIT, Haifa, Israel*

[4] *Department of Pediatrics A, Rambam Healthcare Campus, Haifa, Israel*

**Editors:** Accepted for publication at MIDL 2026

## Abstract

Estimating multi-component $T_2$ relaxation distribution from Multi-Echo Spin Echo (MESE) MRI is a severely ill-posed inverse problem, traditionally approached with regularized non-negative least squares (NNLS). In abdominal imaging, and in the pancreas in particular, low Signal-to-Noise Ratio (SNR), and residual uncorrelated noise between reconstructed echoes challenge both classical solvers and deterministic deep learning models. We introduce a **bootstrap-based inference framework for robust distributional $T_2$ estimation**, which performs stochastic resampling of the echo train and aggregates predictions across multiple echo subsets. This strategy treats the acquisition as a distribution rather than a fixed input, yielding **variance-reduced, physically consistent** estimates and converting deterministic relaxometry networks into **probabilistic ensemble predictors**. Building on the P2T2 architecture, our method applies inference-time bootstrapping to smooth residual noise artifacts, increase tolerance to stochastic inference errors, and enhance fidelity to the underlying relaxation distribution.

We demonstrate a clinical application of the proposed approach for functional and physical assessment of the pancreas. Currently available techniques for noninvasive pancreatic evaluation are limited due to the organ's concealed retroperitoneal location and the procedural risks associated with biopsy, driven in part by the high concentration of proteases that can leak and cause intra-abdominal infection. These constraints highlight the need for functional imaging biomarkers capable of capturing early pathophysiological changes. A prominent example is type 1 diabetes (T1DM), in which progressive destruction of beta cells begins years before overt hyperglycemia, yet no existing imaging modality can assess early inflammation or the decline of pancreatic islets. A further unmet need lies in characterizing pancreatic lesions suspected of malignancy: although malignant and benign lesions differ in their physical properties, current imaging methods do not reliably distinguish between them.

To examine the clinical utility of our method, we evaluate performance in **test–retest reproducibility study** ($N = 7$) and a **T1DM versus healthy differentiation task** ($N = 8$). The proposed approach achieves the lowest Wasserstein distances across repeated scans and demonstrates superior sensitivity to subtle, physiology-driven shifts in the relaxation-time distribution, outperforming classical NNLS and non-bootstrapped deep learning baselines. These results establish inference-time bootstrapping as an effective and practical enhancement for quantitative $T_2$ relaxometry in low-SNR abdominal imaging, enabling more stable and discriminative estimation of relaxation-time distributions.

**Keywords:** Multi-component T2, Quantitative MRI, Physics-informed Neural Networks, Bootstrapped Inference, Test-Retest Reliability, Microstructural Biomarkers

## 1. Introduction

Type 1 Diabetes Mellitus (T1DM) is a chronic autoimmune disease characterized by progressive loss of insulin-producing beta cells in the pancreas. Current diagnostic tools, serological assays and blood glucose measurements, typically identify the disease only after substantial beta-cell destruction has occurred. This motivates the development of non-invasive imaging biomarkers capable of detecting early microstructural alterations, such as inflammation or edema, that precede overt hyperglycemia (Bonner Weir et al., 1983; Jacobsen et al., 2020).

Quantitative Magnetic Resonance Imaging (qMRI), and multi-component $T_2$ relaxometry, in particular, provide a powerful framework for probing such microscopic changes. Unlike conventional $T_2$-weighted imaging, which is influenced by acquisition settings and subjective image interpretation, multi-component relaxometry seeks to recover a *distribution* of relaxation times that reflects different water environments within tissue (Radunsky et al., 2021; Fatemi et al., 2020; Margaret Cheng et al., 2012). The MESE signal can be expressed as a Fredholm integral equation of the first kind:

$$y(TE) = \int K(TE, T_2), p(T_2), dT_2 + \eta, \tag{1}$$

where $K$ is the decay kernel, commonly computed via the Extended Phase Graph (EPG) formalism to account for $B_1$ inhomogeneities (Hennig, 1988), $p(T_2)$ is the relaxation-time distribution, and $\eta$ represents measurement noise.

Recovering $p(T_2)$ from discrete MESE measurements is a highly ill-posed inverse problem. Classical methods such as Tikhonov-regularized Non-Negative Least Squares (NNLS) (Doucette et al., 2020; Canales-Rodríguez et al., 2021; Bai et al., 2014; Chatterjee et al., 2018) often yield unstable or overly sparse solutions, especially in abdominal imaging where low SNR and residual reconstruction noise disrupt individual echoes. Deep learning approaches, including Model-Informed Machine Learning (MIML) (Yu et al., 2021), accelerate estimation but remain sensitive to noise outliers and may fail to generalize across variable MESE protocols.

In this work, we introduce a **bootstrap-based inference strategy** that transforms deterministic $T_2$ relaxometry networks into robust probabilistic estimators. At inference time, the method stochastically resamples subsets of the echo train and aggregates the resulting predictions, effectively averaging out residual uncorrelated noise and stochastic prediction errors, thereby reducing variance in the recovered distribution $p(T_2)$. This ensemble view treats the acquisition as a distribution rather than a fixed signal, markedly improving the robustness to low-SNR conditions.

Our approach is based on the Physically Primed T2 (P2T2) architecture (Ben-Atya and Freiman, 2023), which encodes the echo-time schedule directly into the network, allowing generalization across heterogeneous MESE protocols. Although P2T2 mitigates protocol dependence, its deterministic formulation remains vulnerable to aleatoric noise; inference-time bootstrapping resolves this limitation and enhances sensitivity to subtle changes driven by physiology in pancreatic tissue.

In two experimental settings, a **test–retest reproducibility study** ($N = 7$) and a **T1DM versus healthy differentiation task** ($N = 8$), our method achieved the lowest Wasserstein distances and produced the largest distributional separations between-groups. These results demonstrate that bootstrap-enhanced inference substantially improves stability and discriminative power in quantitative $T_2$ relaxometry for low-SNR abdominal imaging.

**Contributions** This work makes the following key contributions:

- We introduce an **inference-time bootstrap strategy** for stabilizing multi-component $T_2$ distribution estimation. By resampling echo subsets and aggregating ensemble predictions, the method reduces variance and increases robustness to low-SNR, residual uncorrelated noise and stochastic inference errors.

- We perform a **comprehensive statistical evaluation** using AUC, Hellinger distance, and the Kolmogorov–Smirnov p-value, demonstrating that bootstrap-enhanced inference yields more reproducible and more discriminative $T_2$-derived biomarkers than existing classical and deep learning baselines.

## 2. Methods

### 2.1. The Inverse Problem and Forward Model

Estimating the $T_2$ relaxation distribution from multi-echo data can be formulated as an inverse problem, in which a physical signal-decay model is fitted to the acquired MRI echo train. Given multi-echo spin-echo (MESE) data sampled at echo times $TE_{i=1}^{N}$, the objective is to infer a discretized distribution $p(T_2)$ that explains the measured signal $\mathbf{s}$.

This is commonly posed as a regularized least-squares optimization:

$$\widehat{p(T_2)} = \arg \min_{p(T_2)} \sum_{i=1}^{N} |s(TE_i) - s_i|_2^2 + \lambda \mathcal{R}, \tag{2}$$

where $s_i$ is the measured signal, $s(TE_i)$ is the forward-model prediction, and $\mathcal{R}$ enforces smoothness and physical plausibility on the estimated relaxation distribution.

The MRI signal is modeled using the Extended Phase Graph (EPG) formalism (Hennig, 1988; Weigel, 2015). The predicted signal is:

$$s(TE_i) = \sum EPG(TE_i, T_1, T_2, \alpha) \cdot p(T_2) \tag{3}$$

where $\alpha$ denotes the refocusing flip angle train. Thus, the goal is to learn the inverse mapping $f : (\mathbf{s}, \mathbf{TE}) \to p(T_2)$

Figure 1: **Representative $T_2$ distribution estimation workflow (Control Subject).** **Left:** Axial MRI slice (TE=77.4 ms) with the pancreatic ROI centroid indicated. **Center:** Signal decay analysis: The model accurately fits the physical decay curve ($R^2 = 0.986$), matching the predicted signal (line) to the observed echoes (dots). **Right:** Regional $T_2$ distribution analysis. Faint colored curves show the final bootstrapped $T_2$ distributions for individual pixels within the ROI, illustrating local spatial heterogeneity. The solid black curve denotes the spatially averaged distribution across the ROI. Dashed lines depict the Gaussian decomposition of this average into Short-$T_2$ (red) and Long-$T_2$ (blue) components.

## 2.2. Bootstrapped Physically-Primed Networks for Robust $T_2$ Estimation

Our main contribution is a bootstrap-based reconstruction framework that converts a deterministic $T_2$ estimator into a robust, ensemble-driven model. Instead of relying on the full echo train, the method repeatedly resamples small subsets of echoes and aggregates the resulting predictions. We hypothesize that this approach addresses two distinct sources of variability inherent to the low-SNR regime: (1) residual uncorrelated noise between reconstructed echo images, which originates from thermal noise in each radial spoke acquisition and remains partially uncorrelated between echoes despite the shared timing; and (2) occasional incorrect estimation of the $T_2$ distribution by the neural network, where inference-related errors are effectively uncorrelated across independent resampling instances. By aggregating predictions, the method effectively marginalizes over these error sources.

Given an echo train of length $N_{\text{total}}$, the method proceeds as follows:

1. **Subset Resampling:** For each bootstrap iteration $b = 1, \ldots, B$, we construct a subset of $m < N_{\text{total}}$ echoes by always including the first echo and randomly sampling the remaining $m - 1$ echoes, yielding reduced signal and TE vectors.

2. **Prediction:** Each subset is processed by a physically-primed model trained for inputs of length $m$, producing a distribution estimate $\hat{p}_b(T_2)$.

3. **Ensemble Aggregation:** The final $T_2$ distribution is obtained by averaging bootstrap predictions:

$$\hat{p}_{\text{final}}(T_2) = \frac{1}{B} \sum_{b=1}^{B} \hat{p}_b(T_2).$$

We set $B = 200$ throughout all experiments, which provides an effective balance between computational cost and variance reduction. This ensemble mechanism significantly improves reconstruction stability in low-SNR conditions by smoothing out stochastic variability in both the signal and the inference process.

### 2.3. Backbone Architecture

As the underlying predictor, we employ a fully connected network that jointly receives the MRI signal and its corresponding echo-time (TE) schedule. For each voxel, the input vector is defined as

$$\mathbf{x} = [\,\mathbf{s} \oplus \mathbf{TE}\,],$$

where $\mathbf{s}$ denotes the measured signal and $\mathbf{TE}$ the physical echo times. Explicitly encoding TE values—rather than echo indices—allows the model to learn protocol-invariant decay relationships and generalize across heterogeneous acquisition designs.

The backbone consists of 12 fully connected layers with 256 units per layer and ReLU activations. The final layer is a SoftMax classifier over a discretized grid of $T_2$ values, yielding a normalized relaxation-time distribution. Because the network is physically primed via the TE schedule, it is inherently protocol-agnostic and supports the bootstrap strategy without requiring per-scanner or per-protocol model retraining.

## 3. Experimental Setup

### 3.1. Participants and Study Design

Two cohorts were analyzed to evaluate algorithmic stability and clinical sensitivity:

- **Test–Retest Reliability:** Seven healthy volunteers were recruited. Each subject was scanned twice within an interval of approximately 10 minutes to evaluate the repeatability of the estimator in the absence of physiological intervention.

- **Glucose Challenge:** A cohort of eight subjects, comprising four patients with T1DM and four healthy controls. Participants were scanned at two timepoints: baseline (pre-glucose), and after a 30-minute wait (post-glucose intake), to probe metabolic-induced microstructural changes.

All participants provided informed consent, adhering to institutional guidelines.

### 3.2. Image Acquisition and Segmentation

Abdominal MRI was acquired using a Multi-Echo Spin Echo (MESE) sequence on a 3T scanner. The protocol utilized a train of 32 echoes to sample the decay curve.

The pancreas was manually segmented on anatomical images and subdivided into body, and tail regions. These segmentation masks were subsequently resampled and projected onto the co-registered ME-T2W space.

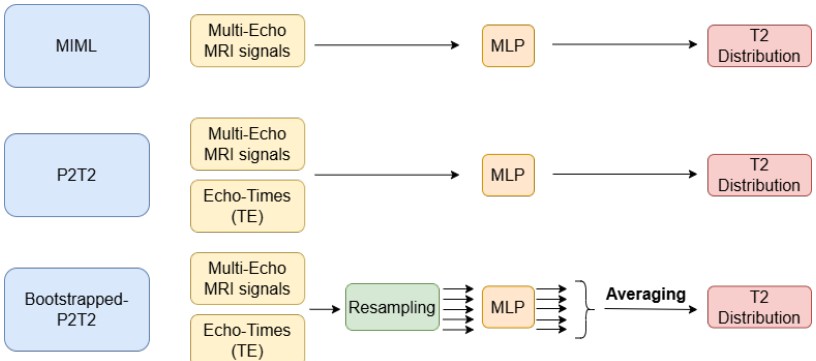

Figure 2: Comparison of the evaluated model architectures. The baseline MIML directly maps signals to $T_2$ distributions; P2T2 incorporates the explicit TE encoding; and the proposed Bootstrapped-P2T2 adds inference-time resampling and ensemble averaging to improve robustness and stability.

### 3.3. T2 Distribution Modeling

Figure 2 presents a schematic comparison of the evaluated model architectures. In this study, we evaluated three architectures:

1. **MIML (Yu et al., 2021):** A baseline neural network that maps the input signal directly to a $T_2$ distribution without encoding the echo-time schedule.

2. $P_2T_2$ (**Ben-Atya and Freiman, 2023**): An extension of the MIML architecture that concatenates the specific TE acquisition array with the input echo signal, allowing the model to generalize across different protocols.

3. **Bootstrapped-$P_2T_2$:** Our proposed robust framework, which performs repeated $P_2T_2$ estimation on random echo subsets (using $m = 14$, a configuration validated for stability in our sensitivity analysis).

**Training Data.** To ensure a fair comparison, **all evaluated models (including the MIML baseline) were trained from scratch on the same** synthetic MESE datasets generated via the Extended Phase Graph (EPG) formalism to simulate a wide range of biologically plausible water pools. Specifically, we simulated a mixture of Short ($< 30ms$), Medium ($50 - 300ms$), and Long ($> 300ms$) $T_2$ components to cover the biophysical range of pancreatic parenchyma, fibrosis, and fluid/edema. Each voxel-wise prediction is a discretized probability density function $p_v(T_2)$.

**Biomarker Extraction.** To quantify the differences between the inferred distributions, we utilized the Wasserstein distance ($W_1$) as our primary scalar biomarker. Unlike simple summary statistics (e.g., mean or median), $W_1$ captures the global geometry of the distribution, measuring the minimum "cost" required to transform one distribution into another. For each anatomical ROI, we computed the $W_1$ metric between the inferred distributions from paired scans (e.g., test vs. retest or pre- vs. post-glucose), effectively condensing the

complex distributional output into a single, interpretable distance value for downstream analysis.

**Statistical Analysis.** We applied statistical evaluation on the extracted Wasserstein biomarkers to assess two key properties: model stability in the test-retest scenarios and discriminative power in the T1DM group analysis. Given the limited size of our dataset and the non-normal distribution of the biomarkers observed in the violin plots, we avoided parametric assumptions. Instead, we utilized robust non-parametric metrics—specifically the Area Under the Curve (AUC), Hellinger distance, and the Kolmogorov-Smirnov (KS) test $p$-value—to quantify the separation capabilities and statistical significance of the differences between T1DM and Control groups.

## 4. Results

### 4.1. Algorithmic Stability (Test-Retest)

Figure 3 summarizes the reproducibility of the evaluated models across repeated scans, quantified using the Wasserstein distance ($W_1$) between distributions computed within identical ROIs of the pancreas body and tail. Lower values indicate higher stability. Across all subjects, the proposed Bootstrapped-P2T2 consistently achieved the lowest median and variance of $W_1$, demonstrating substantially improved robustness in low-SNR abdominal conditions. To benchmark against standard uncertainty quantification methods, we evaluated a Deep Ensemble of MIML networks (N=5). As shown in Figure 3, the MIML Ensemble yielded negligible improvement over the single MIML baseline. This suggests that standard model-uncertainty ensembles are insufficient for mitigating the signal corruption found in abdominal MESE. In contrast, the deterministic P2T2 model improved over both MIML baselines, highlighting the advantage of explicitly encoding the TE schedule; however, its variability remained noticeably higher than that of the bootstrapped ensemble. These results indicate that inference-time stochastic resampling plays a critical role in stabilizing multi-echo $T_2$ estimates, yielding markedly more reproducible distributions in the anatomical regions most relevant to pancreatic endocrine tissue.

### 4.2. Separation Between T1DM and Controls

We next assessed each model's ability to detect physiological changes between the pre- and post-glucose scans. Table 1 summarizes the statistical analysis results. While the baseline MIML achieved the highest AUC (0.823), the proposed Bootstrapped-P2T2 demonstrated superior statistical significance, achieving the lowest KS p-value ($4.79 \times 10^{-4}$) and the largest distributional separation (Hellinger distance = 0.427). This suggests that Bootstrapped-P2T2 provides a more robust differentiation between T1DM and healthy tissue, whereas the high AUC of MIML may be driven by its larger variance rather than distinct physical separation. In contrast, all scalar-based methods—both deterministic and bootstrapped—failed to achieve statistical significance ($p > 0.05$), confirming that multi-component modeling is essential for this task.

Figure 4(a) shows the distribution of Wasserstein distances ($W_1$) between timepoints for each subject in the pancreas body and tail. Bootstrapped-P2T2 provided the clearest group separation, with T1DM subjects exhibiting markedly larger distributional shifts than

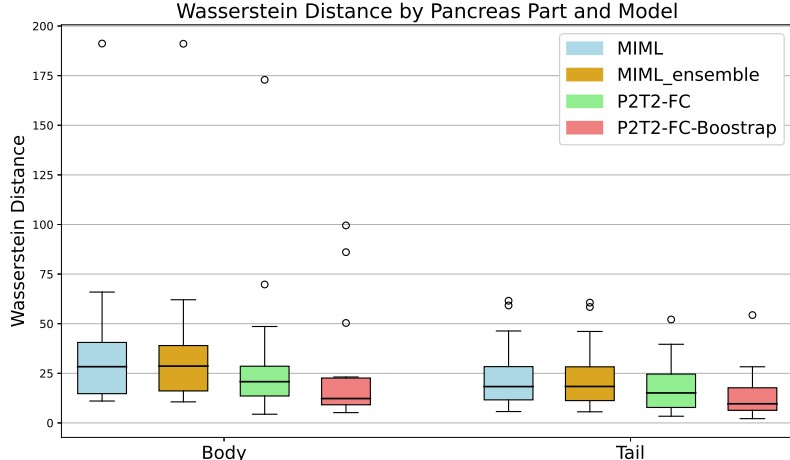

Figure 3: **Test–retest stability across anatomical regions.** Boxplots show the Wasserstein distance ($W_1$) between repeated scans for the pancreatic body and tail. The proposed **Bootstrapped-P2T2** model (red) exhibits consistently lower median error and reduced variance compared with the deterministic P2T2 (green) and baseline MIML (blue), demonstrating improved robustness and repeatability across both regions.

healthy controls and with minimal overlap between groups (Fig. 4a, far right). Deterministic P2T2 captured the same directional trend but with higher overlap (Fig. 4a), reflecting its greater sensitivity to noise. MIML showed statistical differences but substantial overlap between groups (Fig. 4a, left), indicating that much of its variation is not physiologically driven. Classical $T_2$ scalar reconstruction, including $\Delta T_2$ and $\Delta M_0$, showed even higher overlap (Fig. 4b–c), demonstrating their limited ability to capture microstructural alterations. Notably, while applying bootstrapping to the scalar fit reduced variance (Table 1), but it failed to achieve the discriminative power of the multi-component P2T2 approach, highlighting the necessity of the full distributional model. Figure 5 presents two representative subjects, illustrating the markedly larger distributional shift seen in a T1DM participant relative to the minimal shift in a healthy control.

### 4.3. Sensitivity Analysis: Influence of Subset Size

To assess the robustness of the proposed framework to the choice of bootstrap hyperparameters, we performed an ablation study on the subset size $M$. We evaluated the test-retest reliability across a range of subset sizes $M \in \{14, 16, 20, 24\}$, corresponding to resampling fractions of approximately 44% to 75% of the echo train. Figure 6 presents the distribution of Wasserstein distances ($W_1$) for each configuration. The results indicate that the method is highly stable with respect to $M$. The median reproducibility error remained consistently low ($< 15$ ms) across all tested values, with no statistically significant performance degradation observed as $M$ was varied. This stability suggests that the performance gain is primarily driven by the mechanism of stochastic resampling—which effectively averages out

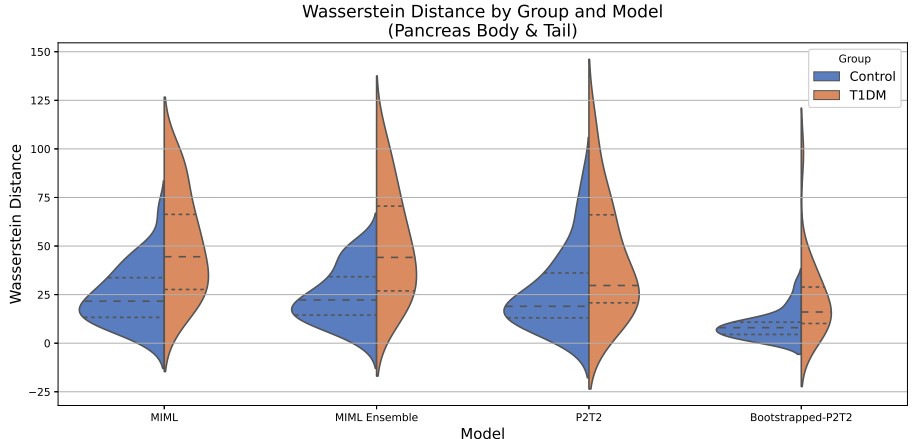

($a$) Distribution-based Models (Wasserstein Distance)

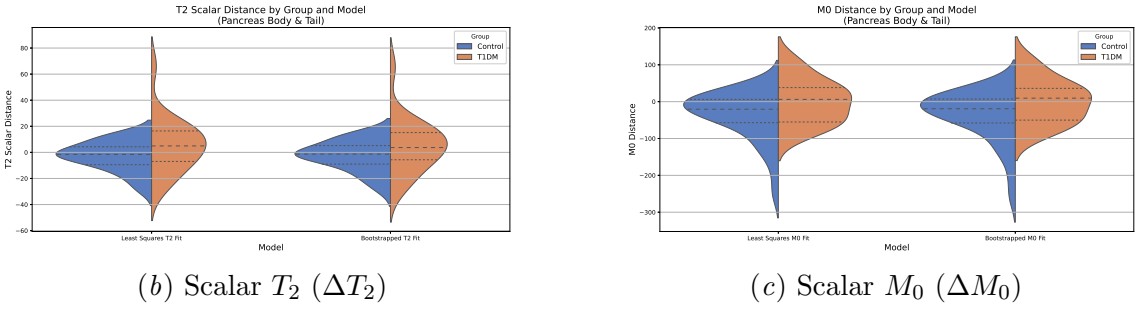

($b$) Scalar $T_2$ ($\Delta T_2$)  ($c$) Scalar $M_0$ ($\Delta M_0$)

Figure 4: **Sensitivity analysis across models (Pancreas Body & Tail).** (a) The combined Wasserstein distance plot compares all distribution-based models. The **Bootstrapped-P2T2** model (far right in a) shows the distinct separation between T1DM (orange) and Controls (blue), reducing the overlap seen in other models. (b–c) Classical scalar metrics ($T_2$ and $M_0$) exhibit larger overlaps between groups, highlighting the improved sensitivity gained from the full-distribution modeling in (a).

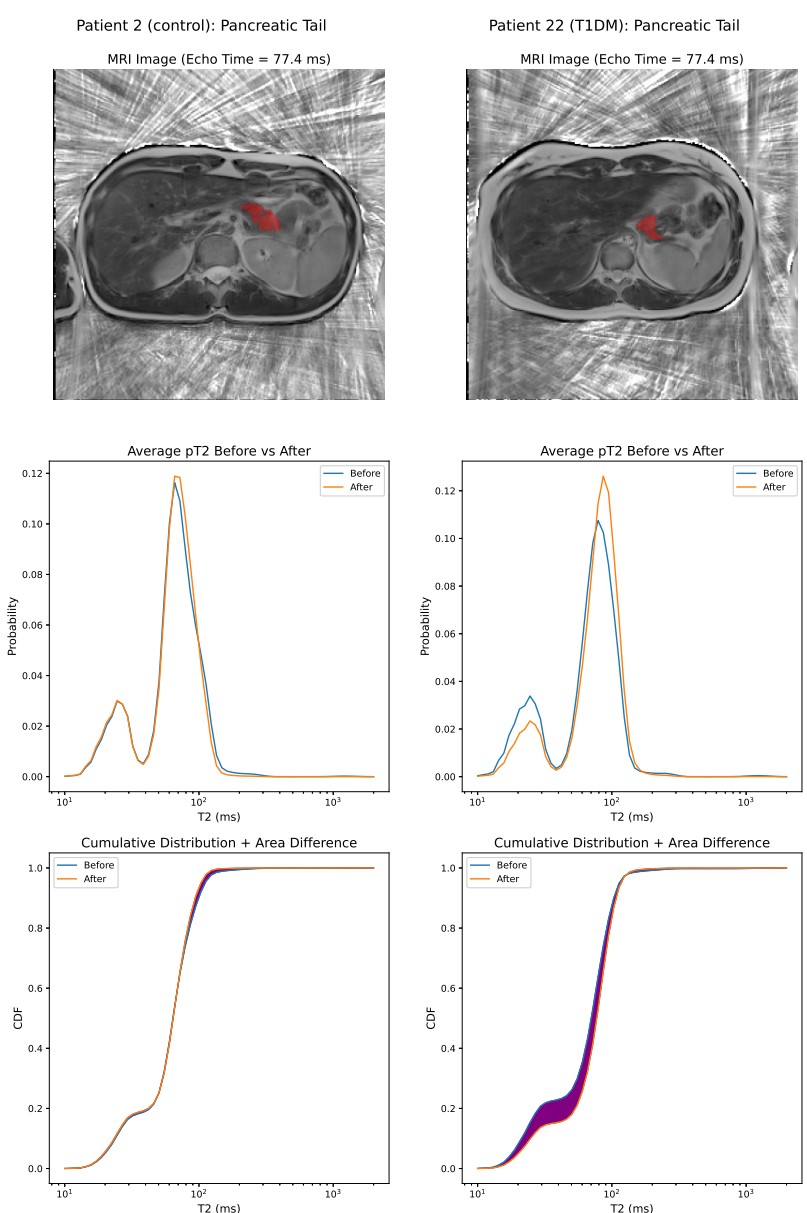

Figure 5: Representative response to glucose challenge in the pancreatic tail, using the Bootstrapped-P2T2. **Left:** Healthy Control subject. **Right:** T1DM subject. **Top Row:** Axial MRI (10th echo) with the pancreatic tail segmentation overlaid in red. **Middle Row:** Average $T_2$ distributions ($\bar{p}(T_2)$) within the ROI before (blue) and after (orange) glucose intake. **Bottom Row:** Cumulative Distribution Functions (CDFs) highlighting the gap between timepoints. Note the significantly larger distributional shift in the T1DM subject compared to the stable profile of the control, indicating an altered microstructural response to metabolic stress.

| Model / Metric | AUC ↑ | Hellinger ↑ | KS p-value ↓ |
|---|---|---|---|
| MIML | **0.823** | 0.287 | $2.36 \times 10^{-3}$ |
| MIML Ensemble | 0.794 | 0.246 | $8.30 \times 10^{-3}$ |
| P2T2 | 0.693 | 0.237 | $1.14 \times 10^{-1}$ |
| **Bootstrapped-P2T2** | 0.800 | **0.427** | $\mathbf{4.79 \times 10^{-4}}$ |
| Scalar T2 (Least Squares) | 0.629 | 0.244 | $9.95 \times 10^{-2}$ |
| Scalar M0 (Least Squares) | 0.632 | 0.124 | $1.37 \times 10^{-1}$ |
| Scalar T2 (Bootstrapped) | 0.603 | 0.279 | $3.99 \times 10^{-1}$ |
| Scalar M0 (Bootstrapped) | 0.605 | 0.170 | $1.69 \times 10^{-1}$ |

Table 1: Evaluation of separation capability. Bootstrapped-P2T2 achieves the strongest statistical significance (lowest p-value) and separation (Hellinger), outperforming all baselines.

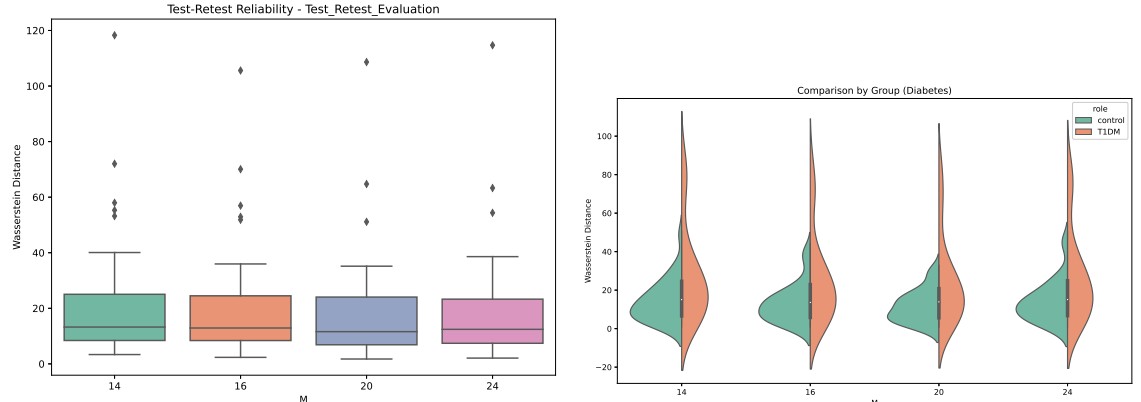

Figure 6: **Sensitivity of Test-Retest Reliability to Bootstrap Subset Size ($M$).** Boxplots showing the Wasserstein distance ($W_1$) between repeated scans for varying subset sizes $M \in \{14, 16, 20, 24\}$. The performance remains stable across all configurations, demonstrating that the method's robustness is not sensitive to small variations in the resampling fraction.

residual uncorrelated noise and sporadic inference errors—rather than the specific tuning of the subset size.

## 5. Discussion

This study demonstrates that inference-time bootstrapping substantially improves the robustness and sensitivity of multi-component $T_2$ relaxometry in abdominal MRI. Across both experiments, test–retest reproducibility and the glucose-challenge paradigm, the proposed Bootstrapped-P2T2 model consistently outperformed deterministic baselines.

In the reproducibility analysis, Bootstrapped-P2T2 achieved the lowest and most stable Wasserstein distances, revealing a clear hierarchy of performance (Bootstrapped-P2T2 > P2T2 > MIML). Crucially, the standard MIML Ensemble failed to improve stability compared to the single MIML baseline. This null result suggests that the dominant error source in abdominal MESE is aleatoric (data-dependent) rather than epistemic (model-dependent). Specifically, our findings indicate that echo-wise stochastic resampling effectively mitigates two key error sources inherent to the radial acquisition: residual uncorrelated image noise and sporadic network prediction errors. This validates our strategy of physical data resampling over standard network ensembling.

In the glucose-challenge cohort, Bootstrapped-P2T2 detected the strongest and most physiologically plausible distributional shifts between pre- and post-glucose scans in the pancreas body and tail, regions known to reflect endocrine tissue function. Deterministic P2T2 captured similar but weaker trends, whereas scalar $T_2$ mapping showed limited discriminative ability even when inference-time bootstrapping was applied to the scalar fit. This confirms that while bootstrapping reduces variance, the multi-component nature of the P2T2 backbone is essential for capturing the subtle microstructural heterogeneity of the pancreas.

While this work is limited by modest sample sizes, our sensitivity analysis demonstrated that the method is highly stable across a range of subset sizes ($M$), mitigating concerns about hyperparameter fragility. Taken together, these findings position Bootstrapped-P2T2 as a robust and biologically meaningful tool for quantitative relaxometry, with potential relevance for developing non-invasive markers of early pancreatic dysfunction in T1DM.

## 6. Conclusion

We presented a novel **Bootstrapped-P2T2 framework** for multi-component $T_2$ relaxometry that integrates physics-informed neural networks with inference-time bootstrapping to enhance robustness and distributional fidelity. This approach improves test–retest reproducibility and provides the clearest separation between T1DM and healthy pancreatic tissue among the evaluated methods. By targeting residual uncorrelated noise and stochastic inference instability through physical resampling, Bootstrapped-P2T2 directly addresses the limitations of low-SNR abdominal imaging. Because no reliable noninvasive tools currently exist for early functional assessment of the pancreas, the ability to recover stable, physiologically meaningful relaxation-time distributions remains a critical unmet need. Overall, the method represents a promising step toward **clinically actionable quantitative pancreatic MRI**, with potential applications in early T1DM detection and pancreatic lesion characterization.

## Acknowledgments

This study was supported in part by a research grant from the Technion's EVPR Foundation for Collaborative Research with the Rambam Healthcare Campus Researchers.

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

## Appendix A. Appendix

### A.1. Acquisition Parameters

Table 2 details the specific acquisition parameters used for the Multi-Echo Spin Echo (MESE) sequence. As requested by the reviewers, we provide the resolution, timing, and acceleration settings used to handle the abdominal imaging constraints.

| Parameter | Value |
|---|---|
| Scanner Field Strength | 3.0 T |
| Sequence Type | Radial Multi-Echo Spin Echo (MESE) |
| Number of Echoes | 32 |
| Echo Spacing ($\Delta$TE) | 7.9 ms (Test-retest) / 7.74 ms (Diabetes analysis) |
| Field of View (FOV) | $360 \times 360$ mm |
| Acquisition Matrix | $208 \times 208$ |
| In-plane Resolution | $1.73 \times 1.73$ mm |
| Slice Thickness | 10.0 mm |
| Breath-hold Strategy | End-expiration breath-hold |

Table 2: Detailed acquisition parameters derived from the image metadata and protocol.

### A.2. Training Data Simulation Parameters

To ensure the network generalizes to the wide range of relaxation times found in the pancreas (including parenchyma, fibrosis, edema, and fluid), we simulated training data using the Extended Phase Graph (EPG) formalism. Table 3 lists the component definitions and ranges used to generate the synthetic training dictionary. Note that while labels such

as "Myelin" or "GM" are borrowed from neuroimaging conventions, the $T_2$ ranges they represent (Short, Medium, Long) provide exhaustive coverage of the biophysical properties relevant to abdominal tissue.

| Component Label | Biophysical Proxy | $T_2$ Mean Range (ms) | $T_2$ Std Range (ms) |
|---|---|---|---|
| Myelin | Macromolecular / Fibrosis | $[15, 30]$ | $[0.1, 5]$ |
| Intra-Cellular (IS) | Parenchyma (Short) | $[50, 120]$ | $[0.1, 12]$ |
| Extra-Cellular (ES) | Parenchyma (Long) | $[50, 120]$ | $[0.1, 12]$ |
| Gray Matter (GM) | Edema / Inflammation | $[60, 300]$ | $[0.1, 12]$ |
| Pathology | Severe Edema / Lesions | $[300, 1000]$ | $[0.1, 5]$ |
| CSF | Fluid / Cysts / Ducts | $[1000, 2000]$ | $[0.1, 5]$ |

Table 3: Parameter ranges used for EPG-based synthetic training data generation. The network was trained on random mixtures of these components (using Dirichlet distributions) to learn the inverse mapping for any continuous $T_2$ distribution within the $1 - 2000$ ms range.

