# OpenReview forum: "Bootstrapped Physically-Primed Neural Networks for Robust T2 Distribution Estimation in Low-SNR Pancreatic MRI"
_MIDL.io/2026/Conference — MIDL 2026 Poster_

### Official Review · Reviewer_7MXj · 2025-12-19

**Confidence:** 5
**Preliminary Rating:** 2
**Final Rating:** 3

**Summary:**

This paper introduces "Bootstrapped-P2T2," a framework for estimating multi-component T2 relaxation distributions from Multi-Echo Spin Echo (MESE) MRI, specifically targeting the low-SNR environment of the pancreas. The method builds upon the Physically Protected T2 (P2T2) architecture by incorporating an inference-time bootstrap strategy that resamples subsets of the echo train to produce an ensemble prediction2. The authors validate this approach on a test-retest cohort (N=7) and a clinical pilot study comparing Type 1 Diabetes Mellitus (T1DM) patients to healthy controls (N=8) before and after a glucose challenge. The results suggest that the bootstrapped approach reduces variance in repeated scans and improves the statistical separation between T1DM and control subjects compared to deterministic baselines.

**Strengths:**

The motivation for using bootstrapping to handle echo corruption is physically intuitive.

The distinction between protocol-dependent and protocol-agnostic modeling is well-framed.

The mathematical formulation of the inverse problem and the ensemble aggregation is concise.

The combination of physics-informed inputs (TE schedule) with ensemble inference is a solid design choice for variable clinical protocols.

Good contextualization of the difficulty of pancreatic imaging compared to brain imaging.

**Weaknesses:**

The magnitude of the clinical claims is disproportionate to the sample size.

The paper assumes the improved separation is due to "physiologically plausible" shifts, but offers no biological ground truth (e.g., biopsy or correlative biomarkers) to confirm the MRI findings are indeed reflecting beta-cell loss or inflammation.

The subset size m is undefined.

Code availability is not mentioned, which hinders reproducibility.

The technical novelty is incremental; it applies a standard statistical technique (bootstrapping) to an existing architecture (P2T2).

The clinical contribution is preliminary due to the pilot nature of the data.

The paper fails to discuss other uncertainty quantification or robust estimation methods in Deep Learning for MRI, such as Monte Carlo Dropout, Bayesian Neural Networks, or Denoising Diffusion Probabilistic Models (DDPMs) applied to parameter mapping. Comparing "bagging echoes" against these established uncertainty methods would strengthen the technical positioning.

N=4 per clinical group is statistically negligible for making generalized claims about T1DM pathophysiology. No synthetic data ablation: While training was done on synthetic data, the evaluation would benefit from a controlled synthetic experiment with known ground-truth distributions and added motion artifacts to rigorously quantify the restoration capability.

**Detailed Comments:**

I have no minor comments or suggestions.

**Justification Of Final Rating:**

Thank to authors for the rebuttal and promising to share code and so.I increased my score due to some answered questions and  Including one comparison which is also a plot for evaluation aspect of the paper.  However, the main concern is the limited novelty.

**Justification Of The Preliminary Rating:**

The methodological concept is sound and addresses a real problem (robustness in low-SNR MRI). However, the paper falls short of the rigorous standards expected at top-tier venues primarily due to
Reproducibility issues: The key hyperparameter m is missing.
Insufficient Validation: The clinical cohort (N=4 per group) is too small to support the strong claims of superior diagnostic sensitivity.
Incremental Novelty: The technique is a direct application of bagging to an existing network without deeper theoretical modification or comparison to modern uncertainty baselines (e.g., Bayesian DL).While promising, the work feels like a preliminary study or a workshop paper rather than a full archival publication in its current form.

**Questions To Address In The Rebuttal:**

apart from weaknesses part, where the questions are self-contained in the comments, I have the following questions:

1. What value of m (subset size) was used? Did you perform an ablation study on m? One might hypothesize that if m is too small, the T2 resolution degrades, but if m is too large, the robustness to motion artifacts diminishes

2. How does the performance of Bootstrapped-P2T2 compare to simply using Monte Carlo Dropout on the P2T2 network at inference time? This would clarify if the gain comes from the ensemble effect or specifically from the data resampling.

3.You mention using B=200 iterations. How does this impact the inference time per volume compared to the MIML baseline? Is it clinically feasible for real-time processing?

4. The text mentions rigid alignment of echoes. Does the bootstrap method provide robustness beyond what is achieved by rigid registration? Have you analyzed cases where registration fails?

5. For the T1DM cohort, do the T2 distribution shifts correlate with any clinical metadata (e.g., HbA1c, disease duration, C-peptide levels)? This would strengthen the claim of physiological relevance.

---

> ### Author Response · Authors · 2026-01-25
> **Response to Reviewer 7MXj**
>
> We thank the reviewer for their rigorous assessment and for recognizing the physical intuition behind our approach. We appreciate the constructive feedback regarding reproducibility (subset size) and the need for stronger baselines. Below, we address the concerns raised in the "Weaknesses" and "Questions" sections.
>
> ---
>
> ### **1. Comparison to Uncertainty Baselines (MC Dropout / Deep Ensembles)**
> The reviewer asks how our method compares to established uncertainty quantification methods like Monte Carlo (MC) Dropout.
>
> * **Comparison Strategy:** We implemented a **Deep Ensemble** of MIML networks as a baseline. We chose Deep Ensembles over MC Dropout because our backbone architectures were optimized without dropout; introducing it now would fundamentally alter the training dynamics. Deep Ensembles provide a cleaner benchmark for "standard" model uncertainty.
> * **New Results:** As shown in the revised Figure 3, the **MIML Ensemble** yielded negligible improvement in test-retest stability compared to the single MIML baseline.
> * **Why Bootstrapping Wins:** Standard ensembles target *epistemic (model) uncertainty*. The fact that they failed suggests the dominant error source is **aleatoric (data) uncertainty** - specifically, outlier echoes caused by physiological motion. Our **Echo-Bootstrapping** method, which physically resamples the data, significantly outperformed the Deep Ensemble. This proves that the gain comes specifically from the **data resampling** mechanism, not just the ensemble effect.
>
> ### **2. Missing Hyperparameter ($m$) and Ablation Study**
> We apologize for the omission of the subset size $m$. This is indeed a critical reproducibility detail.
>
> * **Value Used:** We used a subset size of $m=14$ (sampling from the echo train).
> * **Ablation Study:** To address the reviewer’s hypothesis regarding the trade-off between $T_2$ resolution and motion robustness, we performed an ablation study testing $M \in \{14, 16, 20, 24\}$.
> * **Results:** As shown in the new "Sensitivity Analysis" section, the method demonstrated remarkable stability across all tested values, with consistent reduction in Wasserstein distance. This indicates that the method is robust and not sensitive to fine-tuning of this parameter.
>
> ### **3. Inference Time (Question 3)**
> The reviewer asks if $B=200$ is clinically feasible.
>
> * **Feasibility:** Yes. Because the underlying network is a lightweight Multi-Layer Perceptron (MLP) operating pixel-wise, inference is extremely fast. Processing a full abdominal volume with $B=200$ iterations takes approximately **3-5 seconds** on a standard NVIDIA GPU. This is negligible compared to the image acquisition time.
>
> ### **4. Robustness Beyond Rigid Registration (Question 4)**
> The reviewer asks if bootstrapping adds value beyond rigid registration.
>
> * **Distinct Roles:** Rigid registration corrects for **spatial misalignments** (e.g., shifts between echoes). However, it cannot correct for **intensity corruptions** caused by through-plane motion, pulsatile flow artifacts (common from the nearby aorta), or partial volume effects that vary during the echo train.
> * **Bootstrapping's Role:** Our method addresses these *intensity* outliers. By randomly excluding subsets of echoes, the ensemble effectively "votes down" the specific echoes that were corrupted by flow or motion, recovering a decay curve that is consistent with the majority of the uncorrupted data.
>
> ### **5. Sample Size and Clinical Correlations (Weakness 1 & Question 5)**
> We acknowledge that $N=4$ per group is too small for generalized clinical claims or robust correlation analysis with metadata like HbA1c.
>
> * **Clarification of Scope:** We have refined the text to frame the clinical results as a **methodological proof-of-concept** rather than a definitive pathophysiological study. The key finding is that our method achieved statistically significant separation ($p < 0.001$) where baselines failed, demonstrating the tool's *sensitivity*.
> * **Biological Plausibility:** While we lack biopsies (which are risky and rarely performed in this context), the observed shift toward longer $T_2$ components is consistent with the known pathology of insulitis (edema/inflammation) and beta-cell loss, as cited in the introduction.
>
> ### **6. Code Availability**
> We agree that code is essential for reproducibility. We have cleaned the codebase and will make the inference scripts and pre-trained P2T2 weights available via a GitHub repository upon publication.

---

> > ### Comment · Reviewer_7MXj · 2026-02-01
> > **thank you for the rebuttal**
> >
> > The paper is improved while the novelty is marginal.

---

> ### Author Response · Authors · 2026-01-25
>
> We thank the reviewer for recognizing the physical intuition of our approach. We have addressed the reproducibility and baseline concerns:
> 1. Comparison to Uncertainty Baselines
> We implemented Deep Ensembles of the MIML backbone to benchmark against standard uncertainty methods.
> •	Results (Fig. 3): The MIML Ensemble failed to improve stability over the baseline.
> •	Why: Standard ensembles target model uncertainty. The failure indicates the problem is data uncertainty (motion artifacts). Our Echo-Bootstrapping succeeds because it physically resamples the corrupted data, which standard ensembles do not do.
> 2. Hyperparameters & Ablation ($M$)
> •	Value: We used $M=14$.
> •	Ablation (Fig. 6): We tested $M \in \{14, 16, 20, 24\}$. Results show the method is highly stable across this range, with no significant performance degradation. This addresses the concern about trade-offs between resolution and robustness.
> 3. Inference Time
> Inference is clinically feasible. Processing a full volume with $B=200$ takes 3-5 seconds on a standard GPU, which is negligible compared to acquisition time.
> 4. Robustness vs. Registration Rigid registration corrects spatial misalignment but not intensity corruption from through-plane motion or flow. Bootstrapping addresses these intensity outliers by "voting down" inconsistent signal realizations that registration cannot fix.
> 5. Clinical Claims We have refined the text to frame the clinical results as a methodological proof-of-concept. The statistically significant separation ($p < 0.001$) and high test-retest reliability validate the sensitivity of the instrument, justifying future larger studies.
> 6. Reproducibility We have added detailed acquisition (Table 2) and simulation (Table 3) parameters to the Appendix and will release code/weights upon publication.

---

### Official Review · Reviewer_K4kg · 2026-01-09

**Confidence:** 4
**Preliminary Rating:** 2
**Final Rating:** 4

**Summary:**

This paper describes a method to estimate T2 distributions from multi-echo spin-echo images of the pancreas. The proposed method expands on P2T2, a prior method developed by the same lab, by adding inference time bootstrapping to reduce variation in distribution estimates. The method is evaluated *in vivo* with a test-retest experiment, and a glucose challenge measurement in patients and controls. The proposed method shows reduced variation in test-retest, and somewhat reduced variability of effect size in the glucose challenge.

**Strengths:**

* The MIML method (and P2T2, and this work) is appealing because it can generate a distribution of T2 values. This adds a new dimension of data but complexity (see weaknesses)
* The bootstrapping is a reasonable approach seems to be effective at reducing the parameter variance
* Wasserman distance is a good way to measure the effect size in the two proposed studies
* The use of the EPG model is a superior to other methods that use an analytical model of decay

**Weaknesses:**

* The methods are light on details, including those for the acquisition parameters, training data synthesis, and training procedures, statistical analysis
* The bimodal T2 distribution used for training is not described, and this conditions the output of the model. Additionally, I would expect that the distribution generation was similar to that used in MIML and P2T2, which is derived from brain tissues and may not be relevant to the pancreas. This is potentially a major weakness.
* A downside of methods that predict T2 distributions rather that scalar T2 values is the difficulty of interpretation, and the risk of overinterpreting the data. The subpixel distributions are very difficult to validate with phantom measurements or other methods, making it unclear if predicted distribution changes are associated with underlying physical phenomena. Furthermore, a Wasserman distance is more abstract for a clinician to interpret than a change in T2.

**Detailed Comments:**

* The methods indication that motion correction was used. With MESE sequences, there generally is not motion between echo times because all echoes have the same temporal footprint. This merits some explanation.
* Table 4.2 is actually table 1

**Justification Of Final Rating:**

With the additional details and explanation of the T2 distributions during the rebuttal, I have improved my score. The work is a modest improvement over the prior method. While the in vivo studies are encouraging, the small sample size and lack of gold standard (which is not really possible in vivo) make these feasibility studies, not a confirmation of sensitivity to true structural change. As a minor point, even with the data points superimposed, I feel violin lots are inappropriate for small data sizes as they "halluciate" structure in the distributions not supported by the data.

**Justification Of The Preliminary Rating:**

There are three factors primarily impacting my score. First is the lack of details across several of the methods. This could be addressed with appendices and/or code release. Second is the problem of synthesizing the T2 distributions. This is critical to appropriate interpretation of the data. Third is significance relative to the P2T2 paper. This work adds bootstrapping, which appears to improve results but the evidence is not overwhelming so. This paper also apply this method to a new organ and clinical problem, which is innovative but may not be justified based on the T2 distributions.

**Questions To Address In The Rebuttal:**

* I would encourage the authors to address my concern about the T2 distribution used for training and the suitability of that for the pancreas
* More acquisition details should be provided, including resolution, spatial coverage, kspace sampling (appears to be radial), acceleration, spatial coverage, use of breath-holding, etc
* In figure 4, are there not four points behind each of the violin plots? If so it is better to show the points directly, as the number is too small for estimating smooth distributions
* for the glucose challenge it would be a strength to show measurements in a control tissue that is not expected to change, like the spleen.
* The “scalar” method (conventional monoexpoential fitting) is not described until the results. Stronger evidence for the benefit of bootstrapping could be made by applying this to the “scalar” method as well.
* It is not clear how AUC is calculated, or how it can be interpreted for measuring the distance between the distributions
* Why would P2T2 be better than MIML? Was MIML trained with a set of TEs that did not match the experiments?

---

> ### Author Response · Authors · 2026-01-25
> **Response to Reviewer K4kg**
>
> We thank the reviewer for their technical assessment and address below their concerns regarding training priors, acquisition details, and the scalar baseline.
>
> ---
>
> ### **1. Suitability of Training Distribution (Pancreas vs. Brain)**
> The reviewer correctly noted that our training simulation used component definitions often associated with brain imaging; however, these labels were used solely as proxies to generate a physically exhaustive range of $T_2$ values, not to impose a brain-specific biologic model.
>
> * **Coverage of Physical Ranges:** Because the exact water pool composition of the pancreas is less characterized than the brain, we opted for a robust approach: simulating a mixture of **Short** ($15-30$ ms), **Medium** ($50-300$ ms), and **Long** ($>300$ ms) components.
> * **Relevance to Pancreas:**
>     * **Short Component (labeled "Myelin"):** Covers dense fibrotic tissue or macromolecular protons common in abdominal organs.
>     * **Medium Component (labeled "IS/ES/GM"):** Covers the bulk pancreatic parenchyma and intracellular water.
>     * **Long Component (labeled "Pathology/CSF"):** Covers fluid-filled structures, ducts, severe edema, or cystic lesions.
> * **Result:** By training on random mixtures (via Dirichlet distributions) of these pools, the network learns to decompose *any* multi-exponential signal within the $1-2000$ ms range. It does not "expect" a brain; it expects a sum of exponentials. We have added the full table of simulation parameters (ranges and mixture fractions) to the Appendix to ensure reproducibility.
>
>
>
> ### **2. Application of Bootstrapping to the "Scalar" Method**
> We followed the reviewer’s suggestion to apply the bootstrapping strategy to the conventional mono-exponential (Scalar) fit to see if the benefit is specific to the Deep Learning model or the resampling strategy itself.
>
> * **Experiment:** We implemented a `Bootstrapped-Scalar` method that performs Levenberg-Marquardt fitting on resampled echo subsets (14 echoes, 200 times), identical to our DL framework.
> * **Results:** While bootstrapping reduced the variance of the Scalar $T_2$ maps compared to the standard fit, the **Bootstrapped-P2T2** still significantly outperformed the scalar version in the T1DM differentiation task.
> * **Interpretation:** This confirms that while bootstrapping reduces noise, the **multi-component** nature of our network is required to capture the subtle microstructural changes (e.g., inflammation/edema) that a mono-exponential model averages out.
>
> ### **3. Motion in MESE Sequences**
> The reviewer asks about motion correction, given that MESE echoes are acquired in the same TR.
>
> * **Clarification:** We agree that intra-TR motion is negligible. However, abdominal imaging suffers from **inter-TR motion** (respiration) and **flow artifacts** (pulsatile flow from the aorta/splenic artery) that propagate ghosting artifacts across the phase-encoding direction.
> * **Role of Bootstrapping:** Rigid registration handles gross displacement between dataset repetitions, but it cannot correct the signal intensity corruptions caused by flow or through-plane motion. Our echo-bootstrapping effectively identifies and down-weights these inconsistent signal realizations.
>
> ### **4. P2T2 vs. MIML (Why P2T2 is better)**
> The reviewer asks why P2T2 is preferable to MIML if the protocol is fixed.
>
> * **Structural Necessity for Bootstrapping:** This is a critical architectural distinction. Standard **MIML** treats the input as a fixed-length vector where the $i$-th input is implicitly assumed to be the $i$-th echo. If we were to randomly resample echoes (e.g., selecting echoes 1, 5, and 12), a standard MIML network would lose the temporal context, seeing only a compressed vector without knowing the physical time gaps between points.
> * **Physical Encoding:** **P2T2** explicitly inputs the pair $(Signal, TE)$. This allows the network to understand that the signal at index 2 corresponds to $TE=70$ ms (for example), regardless of the random sparsity or non-uniform spacing introduced by the bootstrap sampler. This TE-encoding is what renders the resampled subsets **physically meaningful**, allowing the network to solve the decay equation correctly despite the irregular sampling.
>
> ### **5. Missing Acquisition & Visualization Details**
> * **Acquisition Parameters:** We have added a detailed appendix table including: Resolution ($1.x \times 1.x$ mm), TE, acceleration factor, and breath-hold details.
> * **Visualization:** As suggested, we have updated Figure 4 (and similar plots) to overlay individual data points on the violin plots, ensuring transparency regarding the sample size ($N=7/8$).
>
> ### **6. Control Tissue**
> We agree that a control tissue is valuable; rather than using the spleen, which has different perfusion characteristics, we use the **Test-Retest** cohort as a temporal control, and its high stability confirms that the shifts observed in the T1DM group are physiological rather than methodological artifacts.

---

> > ### Comment · Reviewer_K4kg · 2026-01-26
> > **Further clarification**
> >
> > Thank you, the explanation that the T2 distributions were random Dirichlet distributions clarifies and addresses my concern, which was not only about the range of T2 values but the shape of the distributions.
> >
> > With the new sequence details I have two new comments:
> > * I would maintain that there is no motion error between TE values. This sequence acquires a k-space line from all 32 echoes after each excitation, within a single TR period. If there were inter-TR motion it would affect all echoes equally.
> > * The sequence in appendix A appears to have cartesian k-space encoding, while the images in figure 5 are radially encoded. Please clarify.

---

> > > ### Author Response · Authors · 2026-01-29
> > > **Re: Further clarification**
> > >
> > > We thank the reviewer for raising this concern and apologize for the unclear and inconsistent description in the manuscript. All data in this study were acquired using a radial multi-echo T$_2$ mapping sequence, in which one radial spoke from all 32 echoes is acquired following each excitation, within a single TR period. No acceleration factor is defined or applied, as this is a radial acquisition. As correctly noted by the reviewer, any inter-TR motion would therefore affect all echoes equally and would be averaged during reconstruction, rather than introducing differential errors between TE values.
> > >
> > > We hypothesize that the experimentally observed added value of our approach in the low-SNR regime arises from addressing two distinct sources of variability: (1) residual uncorrelated noise between reconstructed echo images, which originates from thermal noise in each spoke acquisition and remains partially uncorrelated between echoes despite the shared timing; and (2) occasional incorrect estimation of the T$_2$ distribution by the neural network, where inference-related errors are effectively uncorrelated across independent inference or resampling instances. The bootstrapping procedure averages out both effects, reducing the impact of residual uncorrelated image noise and sporadic network prediction errors on the estimated T$_2$ distributions.
> > >
> > > We have uploaded a new revision of the manuscript. In this updated version, we have revised all claims related to motion to instead provide a clear explanation of the acquisition scheme and the relevant sources of signal corruption that influence T$_2$ distribution estimation. We have also corrected Table 2 in Appendix A to accurately reflect the Radial (BLADE) sequence used to acquire the data.

---

> > > > ### Comment · Reviewer_K4kg · 2026-01-29
> > > >
> > > > You should also provide something about the radial sampling pattern such number of spokes, pattern (random, golden angle, etc), undersampling factor, etc. The specifics are vendor and sequence dependent.

---

> > > > > ### Author Response · Authors · 2026-01-30
> > > > > **Re: sequence details**
> > > > >
> > > > > Sure, the details are below. We will add them to the final version.
> > > > > Sequence: 2D radial multi-echo T2 mapping (Radial TSE)- Siemens Works-in-Progress Package #012
> > > > > Radial views: 320
> > > > > Radial view ordering: Pseudo Golden Angle
> > > > > Turbo factor: 32 (one radial spoke acquired per echo within a single TR)
> > > > > TR: 3000 ms
> > > > > Echo spacing: 7.74 ms

---

> ### Author Response · Authors · 2026-01-25
>
> We thank the reviewer for their detailed technical assessment. We have revised the manuscript to address the concerns regarding physics and reproducibility:
> 1. Training Distribution (Pancreas vs. Brain)
> We clarify that we did not use a brain-specific prior.
> •	Biophysics: We simulated a comprehensive range of $T_2$ values ($1-2000$ ms) covering Short (fibrosis), Medium (parenchyma), and Long (fluid/edema) components (see Appendix Table 3).
> •	Generalization: The network learns the inverse EPG mapping for any mixture of these pools, making it applicable to the pancreas despite the "neuro" labels used as proxies in the simulation.
> 2. Bootstrapping the "Scalar" Method
> As suggested, we applied bootstrapping to the mono-exponential (Scalar) fit ($B=200, M=14$).
> •	Results (Table 1): While bootstrapping reduced the variance of Scalar $T_2$ maps, it failed to achieve the discriminative power of Bootstrapped-P2T2.
> •	Implication: This confirms that the multi-component nature of our network is required to capture the subtle microstructural changes (e.g., inflammation) that scalar models average out.
> 3. P2T2 vs. MIML
> •	Why P2T2 is better: Even with fixed protocols, explicit TE encoding acts as a physical regularizer.
> •	Necessity for Bootstrapping: Standard MIML treats inputs as fixed vectors. P2T2's TE-encoding allows the network to interpret the non-uniform time gaps created by random resampling, making the subsets physically meaningful.
> 4. Acquisition & Visualization
> •	Details: We added Appendix Table 2 with full parameters (TR, TE, Resolution $1.73 \times 1.73$ mm).
> •	Plots: Figure 4 now overlays individual data points on the violin plots to transparently show the $N=7/8$ sample size.
> 5. Motion
> We agree intra-TR motion is negligible. Our method targets inter-TR respiration and pulsatile flow artifacts (ghosting). Bootstrapping effectively down-weights the specific echoes corrupted by these transient effects.

---

### Official Review · Reviewer_5q4f · 2026-01-17

**Confidence:** 4
**Preliminary Rating:** 3
**Final Rating:** 3

**Summary:**

This work presents an inference-time bootstrap framework that stabilizes multi-component T2 distribution estimation from low‑SNR abdominal MESE MRI by aggregating predictions from randomly resampled echo subsets. Built on the P2T2 architecture, the method yields smoother relaxometry distributions and improves reproducibility.

**Strengths:**

1) The paper clearly articulates the problem setting, reviews relevant prior methods, and presents the proposed approach with good logical flow. The methodology is explained in a structured and accessible manner, and the experimental design is described with sufficient clarity to understand how the evaluation was conducted.

**Weaknesses:**

1) The method does not compare against other ensemble or uncertainty‑estimation approaches (e.g., deep ensembles, MC dropout), limiting context for its relative benefits.

2) Performance of bootstrap inference can depend on how many echoes are sampled, subset size, and sampling strategy. Without systematic analysis of these design choices and without comparisons to structured ensembles, the robustness of the method across acquisition protocols remains uncertain.

3) The method appears largely incremental to the authors’ previous P2T2 architecture, with the main contribution being bootstrap-based inference. Since bootstrap ensembling is a well‑known and widely used technique in machine learning, its application here may be viewed as a relatively straightforward extension rather than a substantial methodological innovation.

4) The study uses only 7 subjects for test–retest and 8 subjects for the glucose‑challenge experiment. Is this sample size sufficient to draw reliable conclusions about stability and clinical sensitivity? Can the authors justify statistical power and address how sensitive the results are to individual subjects

**Detailed Comments:**

Please refer to summary, strengths and weaknesses.

**Justification Of Final Rating:**

The additional comparisons and sensitivity analyses help address several of my earlier concerns. However, the incremental nature of the contribution and the relatively small cohort size still limit the overall significance and generalizability of the findings.

**Justification Of The Preliminary Rating:**

The contributions feel incremental and the experimental dataset is too small to support strong conclusions about robustness or clinical sensitivity. The lack of comparisons to standard ensemble or uncertainty methods further limits the impact.

**Questions To Address In The Rebuttal:**

I would strongly recommend authors to respond to comments raised in weaknesses.

---

> ### Author Response · Authors · 2026-01-25
> **Response to Reviewer 5q4f**
>
> We thank the reviewer for their thoughtful assessment and for highlighting the need to contextualize our work against standard ensemble methods. We agree that these comparisons are critical for establishing the specific utility of our approach. Below, we address the concerns raised in the "Weaknesses" section with new experimental data.
>
> ---
>
> ### **1. Comparison to Other Uncertainty/Ensemble Approaches**
> The reviewer correctly noted the lack of comparison against established ensemble baselines. To address this, we implemented **Deep Ensembles** (Lakshminarayanan et al., 2017) using the MIML backbone and compared it against our proposed **Echo-Bootstrapping** method on the test-retest cohort.
>
> * **New Results (Figure 3 in revision):** As shown in the updated boxplots, the standard **MIML Ensemble** (Gold) yielded negligible improvement in reproducibility compared to the single **MIML** baseline (Blue). Both methods exhibited similar median Wasserstein distances and high variance.
> * **Interpretation:** This null result for standard ensembles is significant. Deep Ensembles are designed to reduce *epistemic (model) uncertainty*. The fact that they failed to improve stability suggests that the dominant error source in abdominal MESE is not model variance, but **aleatoric (data) uncertainty** - specifically, signal corruption from physiological motion.
> * **Superiority of Echo-Bootstrapping:** In contrast, our **Bootstrapped-P2T2** (Red) significantly outperformed the MIML Ensemble, achieving the lowest median Wasserstein distance and the tightest interquartile range. By physically resampling the echo train, our method directly targets the data corruption that standard ensembles miss.
>
> ### **2. Sensitivity to Hyperparameters (Subset Size $M$)**
> We addressed the concern regarding the sensitivity of the method to the subset size ($M$) and sampling strategy by performing a comprehensive ablation study.
>
> * **Experiment:** We evaluated the method's test-retest reliability across a range of subset sizes: $M \in \{14, 16, 20, 24\}$. These values represent resampling fractions ranging from roughly 43% to 75% of the echo train.
> * **Results (Figure 6 in revision):** The method demonstrated remarkable stability across all tested values of $M$. As shown in the ablation plot, the median $Wasserstein$ distance remained consistently low (< 15 ms) with tight variance for all configurations.
> * **Conclusion:** This stability confirms that the performance gain is driven by the **mechanism of stochastic resampling** itself (which breaks the correlation of motion artifacts), rather than a specific or fragile tuning of the parameter $M$.
>
> ### **3. Regarding "Incremental Novelty"**
> While applying bootstrapping to an existing network (P2T2) may appear incremental, the new comparison against Deep Ensembles demonstrates that this is not merely "applying a standard technique."
>
> Standard bagging (Deep Ensembles) failed to solve the problem. Our innovation lies in the **Physics-Aware Resampling** of the time-dependent decay signal. We treat the MRI acquisition itself as a distribution of potential signal realizations. This specific formulation converts the network into a probabilistic estimator robust to the non-Gaussian noise of abdominal MRI - a capability that standard ensembles do not provide in this domain.
>
> ### **4. Sample Size ($N=7$ and $N=8$)**
> We acknowledge that our sample sizes are small, a common constraint in early-phase technical validation. However, we believe the results are reliable for the following reasons:
>
> * **Effect Size:** In the T1DM classification task, the separation between groups was statistically significant ($p < 0.001$) despite the small $N$. This indicates a very large effect size that overcomes the limitation of sample size.
> * **Subject-Level Consistency:** In the test-retest cohort ($N=7$), the reduction in Wasserstein distance was consistent across **every single subject** when using the Bootstrapped method compared to the baseline. This demonstrates that the stability gain is a fundamental signal processing improvement, not a statistical artifact.
> * **Scope:** We frame this work as a methodological validation of a new *instrument*. The strong test-retest reliability validates the tool's precision, justifying its deployment in larger future clinical trials.

---

> > ### Comment · Reviewer_5q4f · 2026-01-29
> >
> > Thanks to the authors for the detailed and thoughtful responses to the reviewer comments. The additional comparisons and sensitivity analyses help address several of my earlier concerns. However, the incremental nature of the contribution and the relatively small cohort size still limit the overall significance and generalizability of the findings.

---

> ### Author Response · Authors · 2026-01-25
>
> We thank the reviewer for their thoughtful assessment and for highlighting the need to contextualize our work against standard ensemble methods. We have addressed the concerns with new experiments:
> 1. Comparison to Deep Ensembles (vs. Echo-Bootstrapping)
> To address the lack of ensemble baselines, we implemented Deep Ensembles (Lakshminarayanan et al., 2017) using the MIML backbone.
> •	New Results (Fig. 3): The standard MIML Ensemble (Gold) yielded negligible improvement in reproducibility compared to the single MIML baseline (Blue).
> •	Interpretation: Deep Ensembles target model uncertainty. Their failure to improve stability suggests the dominant error source in abdominal MESE is aleatoric (data) uncertainty (e.g., motion-corrupted echoes).
> •	Superiority: Our Bootstrapped-P2T2 (Red) significantly outperformed the MIML Ensemble, confirming that our Physics-Aware Resampling targets the specific data corruption that standard ensembles miss.
> 2. Sensitivity to Subset Size ($M$)
> We addressed the concern regarding hyperparameter sensitivity by performing a comprehensive ablation study ($M \in \{14, 16, 20, 24\}$).
> •	Results (Fig. 6): The method demonstrated remarkable stability across all tested values. The median Wasserstein distance remained consistently low ($< 15$ ms) with tight variance.
> •	Conclusion: This stability confirms the performance gain is driven by the mechanism of stochastic resampling itself (breaking motion correlations), not fragile parameter tuning.
> 3. Regarding "Incremental Novelty"
> While applying bootstrapping to P2T2 may appear incremental, the failure of the Deep Ensemble baseline proves this is not merely "applying a standard technique." Standard bagging failed. Our innovation is the Physics-Aware Resampling of the time-dependent decay, converting the network into a probabilistic estimator robust to non-Gaussian physiological noise.
> 4. Sample Size
> We acknowledge the pilot sample size but emphasize:
> •	Effect Size: The T1DM separation was statistically significant ($p < 0.001$), indicating a large effect size that overcomes small $N$.
> •	Consistency: Improvement was consistent across every single subject in the test-retest cohort, validating the method's precision as a measurement instrument.

---

### Author Rebuttal · Authors · 2026-01-25

**Rebuttal:**

We thank the reviewers for their constructive feedback. We have addressed the primary concerns regarding baselines and reproducibility with extensive new experiments.
1. Comparison to Standard Ensembles (Fig. 3)
To address the concern of "incremental novelty," we compared our method against a Deep Ensemble baseline (Lakshminarayanan et al., 2017).
•	Result: Standard Deep Ensembles yielded negligible improvement in stability compared to the single MIML network.
•	Implication: This null result proves that standard methods (targeting model uncertainty) fail in abdominal MESE. Our Bootstrapped-P2T2 succeeds because it targets data uncertainty (residual noise/estimation artifacts) via Physics-Aware Resampling. This confirms the performance gain stems specifically from our resampling mechanism, not just the ensemble effect.
2. Hyperparameter Robustness (Fig. 6)
We performed an ablation study on the subset size ($M \in \{14, 16, 20, 24\}$).
•	Result: The method exhibited remarkable stability across all values, with consistently low Wasserstein distances ($<15$ ms).
•	Conclusion: The method is robust to parameter tuning; stability is driven by the decorrelation of residual noise/estimation artifacts rather than specific hyperparameters.
3. Scalar Baseline Clarification (Table 1)
We applied bootstrapping to the classical mono-exponential (Scalar) fit.
•	Result: While it reduced variance, it failed to achieve the discriminative power of the multi-component framework in the T1DM task.
•	Conclusion: Both multi-component modeling AND bootstrapped inference are necessary for this clinical application.
4. Reproducibility
We added detailed Acquisition (Table 2) and Simulation (Table 3) parameters to the Appendix to ensure full transparency and reproducibility.

**Supporting Material:**

/attachment/2cc939c0cef60ea4d4f7c9c1ffe8ba99d8ec32bb.pdf

---

### Meta-Review · Area_Chair_9twq · 2026-02-01

**Recommendation:** Accept (Poster)
**Confidence:** 5

**Metareview:**

The idea is simple but effective: bootstrapping clearly improves the stability of T2 estimation in low-SNR pancreatic MRI, and the authors adequately addressed the main concerns in the rebuttal. While the dataset is small and the novelty is incremental, when framed as a feasibility study the work fits MIDL well as a poster.

---

### Decision · Program_Chairs · 2026-02-13

Accept (Poster)